# Temporal Trends of Exposure to Organochlorine Pesticides in the United States: A Population Study from 2005 to 2016

**DOI:** 10.3390/ijerph19073862

**Published:** 2022-03-24

**Authors:** Mengmeng Li, Rui Wang, Chang Su, Jianwen Li, Zhenyu Wu

**Affiliations:** 1Department of Biostatistics, School of Public Health, Key Laboratory of Public Health Safety and Collaborative Innovation Center of Social Risks Governance in Health, Fudan University, Shanghai 200032, China; 20211020117@fudan.edu.cn; 2National Institute for Nutrition and Health, Chinese Center for Disease Control and Prevention, Beijing 102206, China; wangrui@ninh.chinacdc.cn (R.W.); suchang@ninh.chinacdc.cn (C.S.); 3China National Center for Food Safety Risk Assessment, Beijing 100022, China

**Keywords:** temporal trends, organochlorine pesticides, chemical residuals, NHANES, United States

## Abstract

The current study aimed to investigate temporal trends of serum organochlorine pesticide (OCP) concentrations in the general United States population, approximately 30 years after the prohibition of OCP usage, by using National Health and Nutrition Examination Survey data. The least square geometric means and percent change in OCP concentrations were calculated by a survey weighted multiple linear regression model. Over 2005–2016, OCP concentrations showed significant downward temporal trends. Females had substantially higher concentrations of β-Hexachlorocyclohexane (β-HCH), p,p′-DDE and p,p′-DDT, but lower concentrations of Hexachlorobenzene (HCB) and trans-nonachlor. In addition, females had a more rapid decrease in p,p′-DDT levels over time than males. The overall OCP concentrations increased with age, and the two oldest age groups (aged 40–59 and 60+ years) had substantially lower rates of decrease than the younger age groups (aged 12–39 years). Concentrations and declines in OCPs (except for trans-nonachlor) were higher in Mexican Americans than both non-Hispanic Whites and non-Hispanic Blacks. There is a particular need for the ongoing monitoring of these banned chemicals, and measures should be taken to mitigate the exposure of vulnerable populations, including adults aged over 60, Mexican Americans, females for β-HCH, p,p′-DDE and p,p′-DDT, and males for HCB and trans-nonachlor.

## 1. Introduction

Organochlorine pesticides (OCPs) are one of the most important and effective pesticides available, and have been used intensively in agriculture and pest control around the world [1]. They are volatile and can get into the environment in a variety of ways, such as domestic use, emissions from manufacturing plants, disposal of contaminated wastes and agricultural applications [2]. The primary source of exposure to OCPs for the general population is through the diet via the consumption of milk, fish and meat. Moreover, placental transfer and breastfeeding are thought to be the pathways of fetus and infant exposure to OCPs [3].

OCPs are persistent environment contaminants that act as endocrine disruptors and organ system toxicants, which can thus induce pathologies such as cancer and obesity. Mounting evidence indicates that exposure to OCPs elicits harmful effects on human health, such as metabolic syndrome [4], vitamin D deficiency [5], cognitive function [6], and type 2 diabetes [7]. Thus, OCPs were considered as one of the original 12 persistent organic pollutants according to the Stockholm Convention [8], and the United States (U.S.) Environmental Protection Agency restricted and banned the use of most OCPs during the 1970s and 1980s [9]. Although these chemicals are rarely used in the U.S. nowadays, OCPs remain ubiquitously exposed and can also be detected in the environment and human tissues; they are known to be lipophilic, bio-accumulative and resistant to environmental degradation [10,11,12].

Exploring the long-term temporal trends of OCP residue concentrations is valuable in evaluating the impact of regulatory policies regarding the elimination or restriction of the adverse effect of environmental contaminants and finding factors associated with human exposure. Several studies have focused on the temporal trends of OCP concentrations in the environment or in human tissues, such as in the Canadian Arctic atmosphere [13], the lower reaches of the Yangtze river [14], Australian population blood serum [3], Chinese breast milk samples [15], Northern Norwegian women’s blood serum [16], etc. However, temporal trends of OCP concentrations in the general United States population exposure remain limited. Therefore, the current study was designed to investigate the temporal trends of serum OCP residual concentrations approximately 30 years after the prohibition of OCP usage in the U.S. by using National Health and Nutrition Examination Survey (NHANES) data. Additionally, effect modifications by age, gender, and race were also evaluated.

## 2. Materials and Methods

### 2.1. Study Population

NHANES, conducted by the CDC’s National Center for Health Statistics, is designed to collect nationally representative data of the civilian, non-institutionalized, U.S. population. Participants of the survey completed considerable household interviews, physical examinations, medical-history reports and laboratory tests. All participants (guardians for participants < 18 years of age) gave informed written consent. The current study data came from the six latest consecutive cycles of NHANES, spanning 12 years from 2005–2006 to 2015–2016. The number of available NHANES samples in each cycle was 1973, 2070, 2322, 1910, 2157 and 1989, respectively.

### 2.2. NAHNES Sample Weights and Pooling Strategy

NHANES is a multistage, probability sampling survey. The sample weights were created to adjust for complex survey design, survey nonresponse, and post-stratification to ensure that calculated estimates were representative of the U.S. general population. Subjects in this study were selected from an approximately one-third subset of the survey participants, and subsample weights were assigned to those who had sufficient serum for laboratory testing.

The weighted pooled sample design was first adopted in 2005–2006, in which a larger sample volume was produced to reduce the detection limit, costs, and the number of measurements. The pooling strategy, defined according to the demographic characteristics, included sex, age group, and race/ethnicity. Each serum pool comprised 2–8 individual samples.

### 2.3. Exposure Assessment and Target OCPs

Serum samples were stored frozen until they were shipped to the National Center for Environmental Health for testing. The samples were extracted using liquid/liquid extraction, employing an automated Liquid Handling instrument (Gilson 215 Liquid Handler^®^, Gilson, Inc., Middleton, WI, USA) and sample cleanup was obtained by elution (5% DCM in hexane; 10 mL) of the extract through a column containing, from the top, 0.25 g of silica and 1 g of silica/sulfuric acid (33% by weight). Final determination of target analytes was performed by isotope dilution gas chromatography–high-resolution mass spectrometry GC/IDHRMS. Concentrations of target analytes were reported on two different concentration bases: (i) fresh weight basis (pg/g serum), and (ii) lipid weight basis (ng/g lipid). Details of the laboratory methods are publicly available at: https://wwwn.cdc.gov/nchs/nhanes/ (accessed on 20 March 2022). The limit of detection (LOD) of each OCP is summarized in Appendix A.

β-Hexachlorocyclohexane (β-HCH), Hexachlorobenzene (HCB), p,p′-DDE, p,p′-DDT, trans-nonachlor, Mirex, and Oxychlordane were measured in all six survey cycles, while γ-hexachlorocyclohexane (lindane) (γ-HCH) was measured in 2005–2012 and o,p′-DDT was measured in 2005–2008. Among these OCPs, only those with detection frequencies greater than 60% were considered to be the targeted OCPs in the final analyses, and results were presented based on lipid-adjusted concentration (ng/g lipid).

### 2.4. Statistical Analyses

For OCPs concentrations below the LOD, a value equal to the LOD divided by the square root of 2 was imputed. Survey weighted multivariable linear regressions were constructed to calculate the least square geometric mean (LSGM) of OCP concentrations and the percent change in OCP concentrations associated with a unit increase in the survey cycles (per 2-year cycle, ng/g lipid). The outcome variables were each ln-transformed targeted OCP concentrations, due to the skewed distributions of the OCP concentrations to higher values. The covariates were NHANES survey cycles (continuous) and demographic variables, including sex (male and female), age group (12–19, 20–39, 40–59, and 60+ year) and race/ethnicity (MA: Mexican American; NHB: Non-Hispanic Black; NHW: Non-Hispanic White; and OTHER: other than MA, NHB and NHW). To evaluate the influence of these covariates on OCP concentrations, we first established multiple regression models with only the above-mentioned covariates and then examined the possible effect modifications between NHANES survey cycles and (1) age, (2) sex, and (3) race/ethnicity by including a two-way interaction term. Statistical significance of the interaction term indicated an effect modification. Differences in OCP concentrations were explored within the demographic group for the fixed survey cycle by using pairwise comparisons.

All analyses and visualizations were performed in R (version 4.0.4, https://www.r-project.org, accessed on 20 March 2022), and a two-sided *p* value less than 0.05 was considered statistically significant.

## 3. Results

### 3.1. Temporal Trends in Serum OCP Concentrations by Survey Cycles

β-HCH, HCB, p,p′-DDE, p,p′-DDT, and trans-nonachlor were considered in the current study (Table 1), the estimated half-life in humans and log Kow of which are presented in Appendix A. In each survey cycle, p,p′-DDE was the most abundant OCP, with concentrations of 444.53 ng/g lipid, 378.51 ng/g lipid, 322.3 ng/g lipid, 274.43 ng/g lipid, 233.68 ng/g lipid, and 198.97 ng/g lipid from 2005–2006 to 2015–2016, respectively. The concentrations of p,p′-DDE were more than 20 times higher than that of β-HCH, HCB, p,p′-DDT, and trans-nonachlor (Table 2). Between 2005 and 2016, OCP concentrations decreased in the general U.S. population and the general trend was lower concentrations in more recent years. For β-HCH, the concentrations declined significantly from 7.37 ng/g lipid (95% CI: 6.88, 7.89) in 2005–2006 to 3.29 ng/g lipid (95% CI: 3.1,3.5) in 2015–2016. Similar trends were observed for HCB, p,p′-DDE, p,p′-DDT and trans-nonachlor (Table 2, Figure 1, and Appendix A). All the five OCPs encompassing six survey cycles decreased in concentration significantly. For each survey cycle of two years, percent decrease in adjusted concentrations were 14.87% for β-HCH, 1.57% for HCB, 14.85% for p,p′-DDE, 14.17% for p,p′-DDT, and 8.74% for trans-nonachlor (Table 2).

### 3.2. Temporal Trends in Serum OCP Concentrations by Survey Cycles and Sex

In both sexes, significant downward trends of OCP concentrations were observed across survey cycles (Figure 2 and Appendix A). For β-HCH, females had consistently higher levels compared with males for every survey cycle, and results were significant. Similarly, females had significantly higher concentrations of p,p′-DDE between 2013–2014 and 2015–2016, and p,p′-DDT between 2005–2006 and 2009–2010 than that for males. However, when comparisons were conducted for HCB and trans-nonachlor, the opposite results were obtained: females seemed associated with significantly lower concentrations of HCB between 2011–2012 and 2013–2014 and trans-nonachlor across all survey cycles (Table 3 and Appendix A).

Overall, effect modifications between sex and survey cycles were only observed for p,p′-DDT (Appendix A; *p* = 0.034). The result suggested that females (percent change: −15.17%; 95% CI: −16.23%, −14.1%; *p* < 0.001) had a more rapid decrease in p,p′-DDT levels over time than males (percent change: −13.07%; 95% CI: −14.73%, −11.38%; *p* < 0.001). Notably, males and females showed a similar percent decline and these decreases were statistically significant for β-HCH, HCB, p,p′-DDE and trans-nonachlor (Table 4).

### 3.3. Temporal Trends in Serum OCP Concentrations by Survey Cycles and Age

β-HCH, HCB, p,p′-DDE, p,p′-DDT, and trans-nonachlor in all age groups showed a consistent similar downward trend from 2005–2006 to 2015–2016, as in the general U.S. population. Over all survey cycles, age seemed positively associated with the concentration of OCPs. Namely, subjects aged 60+ years had the highest OCP concentrations, moderate concentrations were found in those aged 40–59 years, followed by 20–39 years, with the lowest being found in those aged 12–19 years (Figure 3 and Appendix A). Pairwise comparisons among age groups in each given fixed survey cycle were statistically significant (*p* < 0.001, Table 3 and Appendix A) except for the comparison between 12–19 years and 20–39 years in 2005–2006 for HCB (*p* = 0.378; Appendix A).

Except for HCB (*p* = 0.169; Appendix A), there were distinct effect modifications between age and survey cycles, where the age group of 12–19 years accounted for the largest percent change for p,p′-DDE, −16.32% (95% CI: −17.95%, −14.65%; *p* < 0.001) and the age group of 20–39 years contained the largest percent change for β-HCH, p,p′-DDT, and trans-nonachlor, −19.02% (95% CI: −21.62%, −16.34%; *p* < 0.001), −15.67% (95% CI: −17.68%, −13.61%; *p* < 0.001), and −10.67% (95% CI: −12.33%, −8.97%; *p* < 0.001), respectively (Table 5).

### 3.4. Temporal Trends in Serum OCPs Concentrations by Survey Cycles and Race

The temporal trend among races were similar with that in the general U.S. population for β-HCH, HCB, p,p′-DDE, p,p′-DDT and trans-nonachlor (Figure 4 and Appendix A). Compared with the NHW and NHB groups, MAs had higher concentrations of β-HCH, HCB, p,p′-DDE and p,p′-DDT, and lower concentrations of trans-nonachlor. In the pairwise comparisons between MAs and NHWs, MAs and NHBs were statistically significant across all survey cycles (Table 3 and Appendix A).

The significant effect modifications between race and survey cycles were noted for all five OCPs between 2005–2006 and 2015–2016 (Table 3 and Appendix A). The declining trend of β-HCH, HCB, p,p′-DDE and p,p′-DDT was steeper in MAs, while the declining trend of trans-nonachlor was steeper in NHWs, compared with both MAs and NHBs. In addition, we observed that the percent decreases in OCP concentrations for all races were statistically significant, except for HCB in NHWs (Table 6; percent change = −0.55%; 95% CI: −1.56%, 0.47%; *p* = 0.290).

## 4. Discussion

In the present study, time trend analyses were used to assess the change in OCP residues in the human blood over time by analyzing biomonitoring data from NHANES. The results showed that most of the OCPs remained to be detectable even approximately 30 years after the prohibition of OCP usage. During the study period, clear decreasing temporal trends of serum OCP concentrations for the general U.S. population were observed, as well as the sex, age and race subgroup populations. The significant downward trends of OCPs observed in this study are consistent with the fact that production and usage levels of these OCPs have decreased over time, proving that legislation works. We focused on highly detectable OCPs in the general population, including β-HCH, HCB, p,p′-DDE, p,p′-DDT and trans-nonachlor. Historically, those OCPs were primarily used as pesticides, and were eventually banned due to environmental concern during the 1970s and 1980s [17,18,19,20]. Similar declining concentrations of OCPs were also observed in environmental and animal studies conducted in America [21,22] and human tissues in other countries [3,23].

As reported in most populations worldwide [24,25,26], our study found that p,p′-DDE was the most abundant compound among the target OCPs. It is the main metabolite of DDT [27], and the half-life is estimated to be as long as 8.6 years [28]. Moreover, p,p′-DDE is highly lipophilic, persists for prolonged periods in the environment, and accumulates in the food chain and the tissues of living organisms. These properties, combined with approximately 30 years of massive DDT use worldwide, lead to a rapid increase in p,p′-DDE levels in all living organisms globally. Considering the associations of high serum levels of p,p′-DDE with adverse health effects, such as prostate cancer [29], global DNA methylation [30], chronic kidney disease [31], etc., it is essential to monitor the concentrations of DDE continuously.

Effect modifications by demographic characters are also important in describing the temporal trend of OCPs and identifying vulnerable populations. The present study indicated that the concentrations of the OCPs were significantly different between males and females during most of the survey cycles. The concentration of OCPs in both sexes has remained controversial, with a higher concentration of β-HCH, p,p′-DDE and p,p′-DDT being reported in females [3,32], and HCB and trans-nonachlor being found to be lower in females [33,34,35]. In our study, β-HCH rates were significantly higher in females across all survey cycles. This trend was also apparent for p,p′-DDE from 2013–2014 to 2015–2016, and p,p′-DDT from 2005–2006 to 2009–2010. Conversely, females were significantly associated with lower concentrations of HCB between 2011–2012 and 2013–2014 and trans-nonachlor across all survey cycles. The observed gender difference could be explained by metabolic capacities, occupations and diets [35]. When assessing the effect modification by gender, females’ p,p′-DDT concentrations showed a relatively steeper decreasing trend compared with males, which may relate to parity, breastfeeding and differences in whole body fat content between females and males [26,36].

Although age seemed positively associated with OCP concentration, subjects aged 60+ years had the highest OCP concentrations, with moderate concentrations in those aged 40–59 years, followed by 20–39 years, and lowest in those aged 12–19 years; there were different declining patterns among these age groups during the study periods. The significant declining trend of OCPs was flatter in the older age groups of 40–69 years and 60+ years, compared with the participants aged under 40 years. These findings suggest differing concentration levels for the age groups representing cumulative OCP exposure and higher concentrations could reflect bio-accumulation from historic exposure, which means the oldest age groups received the greatest exposure and therefore had higher serum levels than their younger counterparts [37]. Moreover, the elimination of the use of these chemicals reduced the exposure of the younger individuals. Another likely explanation is that older people are less active and usually overweight with an accumulation of fat. Since these OCPs are strong lipophilic chemicals and primarily store in adipose tissues [38], higher concentrations were thus observed in people of an older age. Furthermore, different decline rates may be owing to the different half-lives of OCPs in different age groups [39].

The effect of modification by race could be described in our study as there being a higher concentration and greater decline rate in MA subjects for β-HCH, HCB, p,p′-DDE and p,p′-DDT compared with NHWs and NHBs. Evidence suggests that MAs were under a heavy burden of OCPs [28,40], both recently and in previous decades. Mexican Americans have lower levels of socioeconomic status than non-Hispanic Whites and Blacks [41]. They reside in the Southwest and work primarily in high-risk, low-social-position occupations such as farming, forestry and fishing [42]. Previous research has shown that residing on a farm or in the South or West increased risk of exposure to OCPs [43]. In addition, there has been a notable increase in the consumption of fats among MAs, leading to a higher rate of overweight observed in MAs [43,44]. Because adipose tissue acts as a reservoir for lipophilic, liposoluble environmental pollutants, OCPs may thus have a greater level in those with obesity [45,46]. Therefore, a speculation could be that racial differences may be related, at least in part, to MAs’ employment, residence in areas with environmental contaminations, or diet. Moreover, the variabilities in rates of decline over time across different races may be partly due to differences in how OCPs are metabolized by different races [47].

To our knowledge, this is the first and largest study to report temporal trends of OCP concentrations in the blood serum of the general U.S. population. Our study enhances the evidence of population-level exposure to OCPs and offers novel insights of variation in human exposure to OCPs over time. In addition, the relatively large sample size provides well-powered analyses, and the relatively long study period provides well-demonstrated OCP exposure trends. Meanwhile, the findings of this study can be generalized to the general U.S. population, due to the consideration of the design and sampling strategies.

Despite its strengths, the present study also has some limitations. First, due to the cross-sectional design of NHANES, we cannot estimate longitudinal changes in OCPs from the same participant, and thereby address the issue of causality. Second, the use of pool sample data leads to the loss of individual-level information. Thus, variables such as occupation, residence, and genetics are not included in the regression model. However, numerous studies have been conducted using NHANES data, which warrants the results of our present study relatively reliable.

## 5. Conclusions

OCPs remained to be detectable even approximately 30 years after the prohibition of OCP usage, although decreasing temporal trends of serum OCP concentrations in the general U.S. population were observed in our study. In addition, adults aged over 60 and Mexican Americans were found to be vulnerable to OCPs, while females were vulnerable to β-HCH, p,p′-DDE and p,p′-DDT, and males were vulnerable to HCB and trans-nonachlor. Given the adverse health effects associated with high concentrations of OCPs, there is a particular need for the ongoing monitoring of these banned chemicals, and measures should be taken to mitigate the exposure of vulnerable populations.

## Figures and Tables

**Figure 1 ijerph-19-03862-f001:**
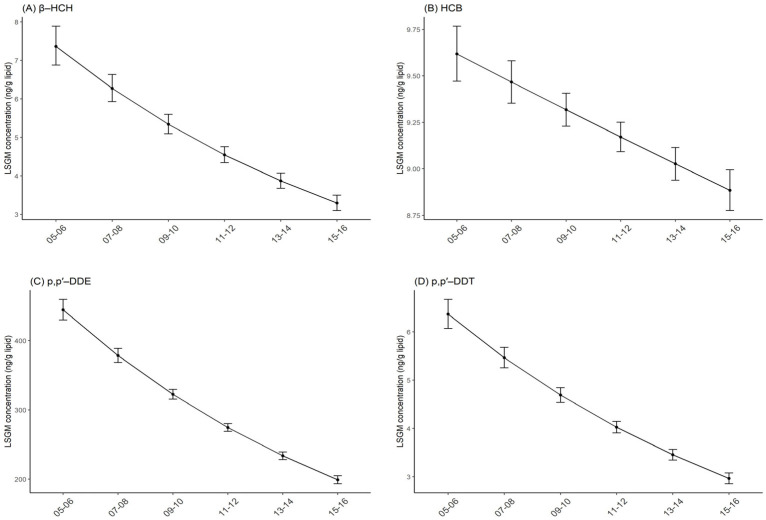
Temporal trends in serum OCP concentrations by survey cycles. Least square geometric mean (LSGM) and 95% confidence interval (95% CI) for each survey cycle estimated from linear regression models adjusted for survey cycles, sex, age and race/ethnicity. Data points represent LSGM, and error bars represent 95% CI. (**A**): β-HCH. (**B**): HCB. (**C**): p,p′-DDE. (**D**): p,p′-DDT.

**Figure 2 ijerph-19-03862-f002:**
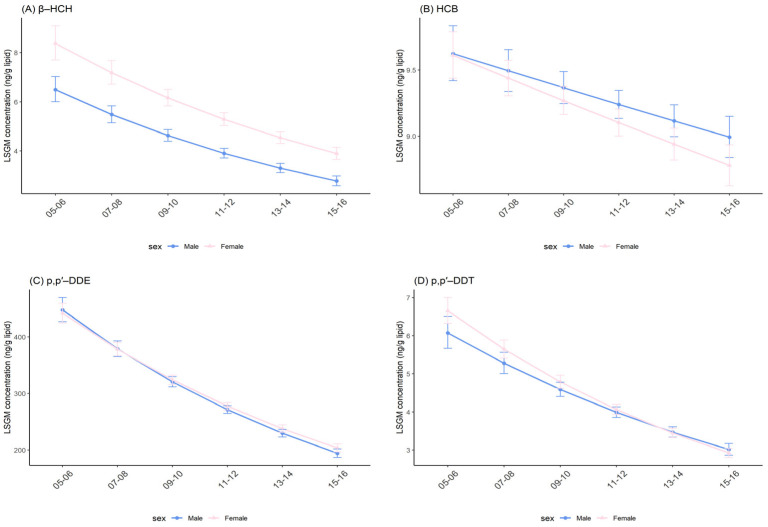
Temporal trends in serum OCP concentrations over survey cycles by sex. Least square geometric mean (LSGM) and 95% confidence interval (95% CI) for each survey cycle estimated from linear regression models adjusted for age group and race/ethnicity; interaction term is survey cycles##sex. Data points represent LSGM, and error bars represent 95% CI. (**A**): β-HCH. (**B**): HCB. (**C**): p,p′-DDE. (**D**): p,p′-DDT.

**Figure 3 ijerph-19-03862-f003:**
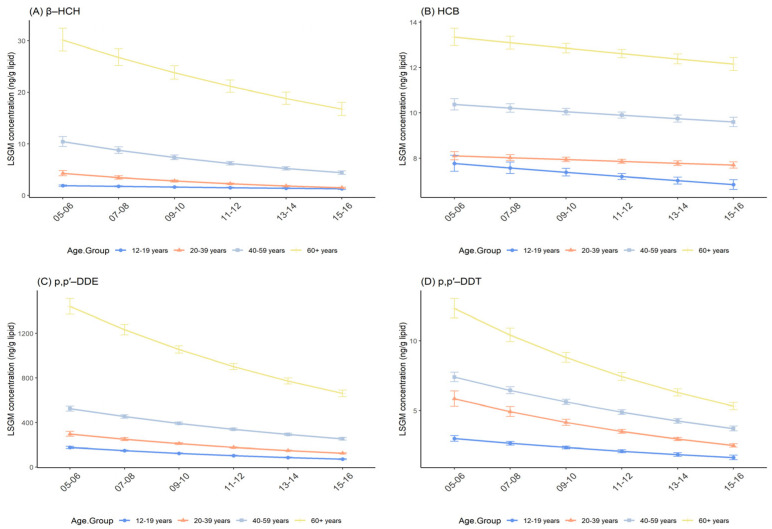
Temporal trends in serum OCP concentrations over survey cycles by age group. Least square geometric mean (LSGM) and 95% confidence interval (95% CI) for each survey cycle estimated from linear regression models adjusted for sex and race/ethnicity; interaction term is survey cycles##age group. Data points represent LSGM, and error bars represent 95% CI. (**A**): β-HCH. (**B**): HCB. (**C**): p,p′-DDE. (**D**): p,p′-DDT.

**Figure 4 ijerph-19-03862-f004:**
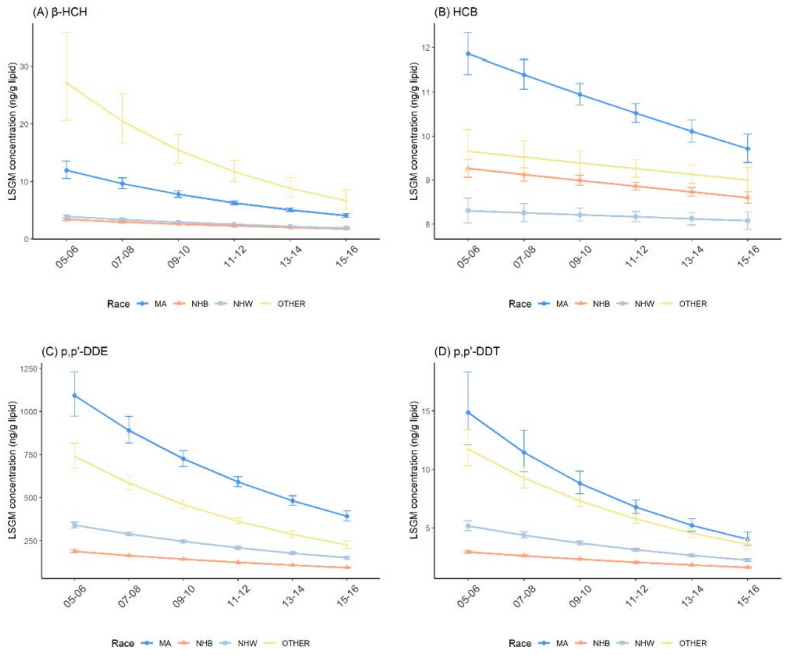
Temporal trends in serum OCP concentrations over survey cycles by race/ethnicity. Least square geometric mean (LSGM) and 95% confidence interval (95% CI) for each survey cycle estimated from linear regression models adjusted for sex and age group; interaction term is survey cycles##race/ethnicity. Data points represent LSGM, and error bars represent 95% CI. (**A**): β-HCH. (**B**): HCB. (**C**): p,p′-DDE. (**D**): p,p′-DDT. MA = Mexican American; NHB = Non-Hispanic Black; NHW = Non-Hispanic White; and OTHER = other than MA, NHB and NHW.

**Table 1 ijerph-19-03862-t001:** Detection frequency (%) of OCPs by Survey Cycles.

Compounds	Survey Cycles					
	2005–2006 (*n* = 1973)	2007–2008 (*n* = 2070)	2009–2010 (*n* = 2322)	2011–2012 (*n* = 1910)	2013–2014 (*n* = 2157)	2015–2016 (*n* = 1989)
β-HCH	75.3	68.9	86.4	82.5	76.1	73.6
HCB	98.8	100.0	100.0	100.0	100.0	100.0
p,p′-DDE	100.0	100.0	99.3	99.6	100.0	97.7
p,p′-DDT	99.2	82.2	96.3	95.6	92.6	85.7
trans-nonachlor	99.6	97.7	99.0	99.2	99.6	99.2
Mirex	59.5	59.5	65.8	65.3	64.1	58.9
Oxychlordane	97.6	94.3	58.5	97.6	98.9	97.4
γ-HCH	0	0	0.3	0.4	/	/
o,p′-DDT	25.1	4.9	/	/	/	/

**Table 2 ijerph-19-03862-t002:** LSGM (ng/g Lipid) by Survey Cycles and Percent Change by the 2-Year Survey Cycle for OCPs.

Compound	Cycles	LSGM (95% CI)	Percent Change ^1^ (95% CI)
β-HCH	2005–2006	7.37 (6.88, 7.89)	−14.87 (−16.42, −13.28)
	2007–2008	6.27 (5.93, 6.63)	
	2009–2010	5.34 (5.09, 5.6)	
	2011–2012	4.55 (4.34, 4.76)	
	2013–2014	3.87 (3.68, 4.07)	
	2015–2016	3.29 (3.1, 3.5)	
HCB	2005–2006	9.62 (9.47, 9.77)	−1.57 (−2, −1.15)
	2007–2008	9.47 (9.35, 9.58)	
	2009–2010	9.32 (9.23, 9.41)	
	2011–2012	9.17 (9.09, 9.25)	
	2013–2014	9.03 (8.94, 9.11)	
	2015–2016	8.88 (8.78, 8.99)	
p,p′-DDE	2005–2006	444.53 (429.88, 459.67)	−14.85 (−15.65, −14.05)
	2007–2008	378.51 (368.55, 388.74)	
	2009–2010	322.3 (315.33, 329.41)	
	2011–2012	274.43 (268.87, 280.11)	
	2013–2014	233.68 (228.31, 239.17)	
	2015–2016	198.97 (193.3, 204.81)	
p,p′-DDT	2005–2006	6.36 (6.07, 6.67)	−14.17 (−15.19, −13.15)
	2007–2008	5.46 (5.25, 5.68)	
	2009–2010	4.69 (4.54, 4.84)	
	2011–2012	4.02 (3.91, 4.15)	
	2013–2014	3.45 (3.35, 3.56)	
	2015–2016	2.96 (2.85, 3.08)	
trans-nonachlor	2005–2006	13.65 (13.13, 14.2)	−8.74 (−9.94, −7.52)
	2007–2008	12.46 (12.12, 12.81)	
	2009–2010	11.37 (11.16, 11.58)	
	2011–2012	10.38 (10.21, 10.54)	
	2013–2014	9.47 (9.25, 9.69)	
	2015–2016	8.64 (8.35, 8.94)	

^1^ Percent change = 100 (e^β^ − 1), where β is the estimated model coefficient for the variable 2-year survey cycle.

**Table 3 ijerph-19-03862-t003:** LSGM (ng/g Lipid) and 95% CI by Survey Cycles and Demographic Groups for β-HCH.

Category	LSGM	95% CI	Pairwise Comparison: *p*		
Sex and Survey (*p* = 0.174)					
Cycle 2005–2006					
Male	6.49	(6, 7.02)			
Female	8.37	(7.7, 9.09)	<0.001		
Cycle 2007–2008					
Male	5.48	(5.15, 5.84)			
Female	7.18	(6.72, 7.68)	<0.001		
Cycle 2009–2010					
Male	4.63	(4.39, 4.88)			
Female	6.16	(5.83, 6.51)	<0.001		
Cycle 2011–2012					
Male	3.91	(3.72, 4.11)			
Female	5.29	(5.03, 5.55)	<0.001		
Cycle 2013–2014					
Male	3.3	(3.11, 3.49)			
Female	4.54	(4.31, 4.78)	<0.001		
Cycle 2015–2016					
Male	2.78	(2.59, 2.99)			
Female	3.89	(3.66, 4.15)	<0.001		
Age Group and Survey (*p* < 0.001)			12–19 years	20–39 years	40–59 years
Cycle 2005–2006					
12–19 years	1.87	(1.74, 2.02)			
20–39 years	4.24	(3.76, 4.78)	<0.001		
40–59 years	10.42	(9.51, 11.43)	<0.001	<0.001	
60+ years	30.14	(28.03, 32.41)	<0.001	<0.001	<0.001
Cycle 2007–2008					
12–19 years	1.73	(1.63, 1.84)			
20–39 years	3.43	(3.13, 3.77)	<0.001		
40–59 years	8.76	(8.14, 9.43)	<0.001	<0.001	
60+ years	26.79	(25.2, 28.47)	<0.001	<0.001	<0.001
Cycle 2009–2010					
12–19 years	1.6	(1.52, 1.69)			
20–39 years	2.78	(2.6, 2.98)	<0.001		
40–59 years	7.37	(6.93, 7.82)	<0.001	<0.001	
60+ years	23.81	(22.54, 25.15)	<0.001	<0.001	<0.001
Cycle 2011–2012					
12–19 years	1.48	(1.4, 1.57)			
20–39 years	2.25	(2.13, 2.38)	<0.001		
40–59 years	6.19	(5.86, 6.54)	<0.001	<0.001	
60+ years	21.16	(20.01, 22.38)	<0.001	<0.001	<0.001
Cycle 2013–2014					
12–19 years	1.37	(1.27, 1.47)			
20–39 years	1.82	(1.72, 1.93)	<0.001		
40–59 years	5.2	(4.9, 5.53)	<0.001	<0.001	
60+ years	18.81	(17.64, 20.05)	<0.001	<0.001	<0.001
Cycle 2015–2016					
12–19 years	1.26	(1.15, 1.39)			
20–39 years	1.48	(1.37, 1.59)	0.021		
40–59 years	4.38	(4.06, 4.72)	<0.001	<0.001	
60+ years	16.72	(15.48, 18.04)	<0.001	<0.001	<0.001
Race * and Survey (*p* < 0.001)			MA	NHB	NHW
Cycle 2005–2006					
MA	11.94	(10.5, 13.57)			
NHB	3.39	(3.18, 3.62)	<0.001		
NHW	3.9	(3.66, 4.16)	<0.001	0.02	
OTHER	27.11	(20.59, 35.69)	<0.001	<0.001	<0.001
Cycle 2007–2008					
MA	9.62	(8.74, 10.6)			
NHB	2.97	(2.83, 3.11)	<0.001		
NHW	3.38	(3.22, 3.54)	<0.001	0.002	
OTHER	20.46	(16.61, 25.2)	<0.001	<0.001	<0.001
Cycle 2009–2010					
MA	7.76	(7.24, 8.32)			
NHB	2.6	(2.52, 2.69)	<0.001		
NHW	2.92	(2.82, 3.03)	<0.001	<0.001	
OTHER	15.45	(13.13, 18.16)	<0.001	<0.001	<0.001
Cycle 2011–2012					
MA	6.26	(5.94, 6.59)			
NHB	2.27	(2.22, 2.33)	<0.001		
NHW	2.53	(2.46, 2.6)	<0.001	<0.001	
OTHER	11.66	(9.97, 13.63)	<0.001	<0.001	<0.001
Cycle 2013–2014					
MA	5.05	(4.77, 5.34)			
NHB	1.99	(1.94, 2.04)	<0.001		
NHW	2.19	(2.12, 2.26)	<0.001	<0.001	
OTHER	8.8	(7.25, 10.69)	<0.001	<0.001	<0.001
Cycle 2015–2016					
MA	4.07	(3.76, 4.4)			
NHB	1.74	(1.68, 1.81)	<0.001		
NHW	1.9	(1.82, 1.98)	<0.001	0.017	
OTHER	6.64	(5.14, 8.59)	<0.001	<0.001	<0.001

* MA: Mexican American; NHB: Non-Hispanic Black; NHW: Non-Hispanic White; OTHER: other than MA, NHB and NHW.

**Table 4 ijerph-19-03862-t004:** Percent Change in Organochlorine Pesticides by the 2-Year Survey Cycle over Sex.

	Compound	Percent Change (95% CI)	*p*-Value
Male	β-HCH	−15.59 (−17.47, −13.66)	<0.001
	HCB	−1.34 (−1.96, −0.73)	<0.001
	p,p′-DDE	−15.37 (−16.55, −14.18)	<0.001
	p,p′-DDT	−13.07 (−14.73, −11.38)	<0.001
	trans-nonachlor	−8.83 (−10.37, −7.26)	<0.001
Female	β-HCH	−14.18 (−15.98, −12.34)	<0.001
	HCB	−1.79 (−2.34, −1.24)	<0.001
	p,p′-DDE	−14.36 (−15.34, −13.36)	<0.001
	p,p′-DDT	−15.17 (−16.23, −14.1)	<0.001
	trans-nonachlor	−8.66 (−9.79, −7.51)	<0.001

**Table 5 ijerph-19-03862-t005:** Percent Change in Organochlorine Pesticides by the 2-Year Survey Cycle over Age Group.

	Compound	Percent Change (95% CI)	*p*-Value
12–19 years	β-HCH	−7.57 (−9.96, −5.11)	<0.001
	HCB	−2.52 (−3.83, −1.2)	<0.001
	p,p′-DDE	−16.32 (−17.95, −14.65)	<0.001
	p,p′-DDT	−11.36 (−13.9, −8.74)	<0.001
	trans-nonachlor	−10.59 (−12.91, −8.2)	<0.001
20–39 years	β-HCH	−19.02 (−21.62, −16.34)	<0.001
	HCB	−1.03 (−1.67, −0.38)	0.002
	p,p′-DDE	−15.82 (−17.46, −14.16)	<0.001
	p,p′-DDT	−15.67 (−17.68, −13.61)	<0.001
	trans-nonachlor	−10.67 (−12.33, −8.97)	<0.001
40–59 years	β-HCH	−15.94 (−18, −13.83)	<0.001
	HCB	−1.54 (−2.28, −0.8)	<0.001
	p,p′-DDE	−13.55 (−14.77, −12.32)	<0.001
	p,p′-DDT	−12.9 (−14.02, −11.77)	<0.001
	trans-nonachlor	−7.38 (−8.97, −5.75)	<0.001
60+ years	β-HCH	−11.12 (−12.92, −9.29)	<0.001
	HCB	−1.85 (−2.7, −1)	<0.001
	p,p′-DDE	−14.44 (−15.66, −13.2)	<0.001
	p,p′-DDT	−15.47 (−16.68, −14.23)	<0.001
	trans-nonachlor	−6.76 (−8.29, −5.21)	<0.001

**Table 6 ijerph-19-03862-t006:** Percent Change in Organochlorine Pesticides by the 2-Year Survey Cycle over Race.

	Compound	Percent Change (95% CI)	*p*-Value
MA *	β-HCH	−19.37 (−22.19, −16.45)	<0.001
	HCB	−3.9 (−5.07, −2.72)	<0.001
	p,p′-DDE	−18.52 (−21.19, −15.76)	<0.001
	p,p′-DDT	−23.09 (−27.59, −18.3)	<0.001
	trans-nonachlor	−8.46 (−10.16, −6.74)	<0.001
NHB *	β-HCH	−12.49 (−14.05, −10.9)	<0.001
	HCB	−1.47 (−2.08, −0.86)	<0.001
	p,p′-DDE	−13.05 (−14.03, −12.06)	<0.001
	p,p′-DDT	−11.29 (−12.33, −10.24)	<0.001
	trans-nonachlor	−8.08 (−9.76, −6.37)	<0.001
NHW *	β-HCH	−13.41 (−14.96, −11.83)	<0.001
	HCB	−0.55 (−1.56, 0.47)	0.290
	p,p′-DDE	−15.04 (−16.36, −13.7)	<0.001
	p,p′-DDT	−15.4 (−17.25, −13.51)	<0.001
	trans-nonachlor	−9.22 (−10.46, −7.96)	<0.001
OTHER *	β-HCH	−24.52 (−30.82, −17.64)	<0.001
	HCB	−1.39 (−2.72, −0.05)	0.042
	p,p′-DDE	−21.11 (−23.78, −18.36)	<0.001
	p,p′-DDT	−21.12 (−24.46, −17.64)	<0.001
	trans-nonachlor	−11.86 (−13.05, −10.65)	<0.001

* MA: Mexican American; NHB: Non-Hispanic Black; NHW: Non-Hispanic White; OTHER: other than MA, NHB and NHW.

## Data Availability

Publicly available datasets were analyzed in this study. This data can be found here: https://wwwn.cdc.gov/nchs/nhanes/ (accessed on 20 March 2022).

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
