# Peer review of "Temporal Trends of Exposure to Organochlorine Pesticides in the United States: A Population Study from 2005 to 2016"

_ijerph, 2022, doi:10.3390/ijerph19073862_

Round 1

Reviewer 1 Report

My opinion was included in the uploaded file.

Reviewer 2 Report

The manuscript as a whole is well written and it contains interesting and publishable data. However, it needs some edits to improve its quality before publication.

Major comments:

-        These OCP contaminants are also known as endocrine disruptors, which can thus induce pathologies such as cancer and obesity. It would be interesting to add this information in the introduction section-        In the materials and methods section, sample preparation and analysis of OCPs were not well described and need to be a bit more detailed. For example, in the result section (line 118) the concentrations were expressed per gram of what? A few lines after it is indicated by gram of lipids. This need to be clearly specified in the materials and methods section.The fate in the environment of these OCPs and their biodegradation and bioaccumulation in food diet and in humans depend on their physico-chemical characteristics, such as their half-life time and their lipophilic. It would be interesting to add an additional table (supplement table) comparing their half-life times and their log Kow.

  • In the result section, the authors reported that subjects aged 60+ years had the highest OCPs concentrations, moderate concentrations in 40-59 years, followed by 20-39 years and lowest in 12-19 years. They suggested that this difference could reflect bioaccumulation from historic exposure, which means the oldest age groups, received greatest exposure and therefore had higher serum levels than their younger counterparts. On the other hand, the elimination of the use of these chemicals had reduced the exposure in the younger individuals. Furthermore, different decline rates may be owing to the different half-lives of OCPs in different age groups. Older people are often less active and have more of a problem of being overweight with an accumulation of fat. Since these OCPs are lipophilic and accumulate in fat in humans, this could explain this difference.
  • The authors reported also that concentrations and declines of OCPs (except for trans-nonachlor) were higher in Mexican American (MAs) than both non-Hispanic White and non-Hispanic Black. This could also be due to the much higher fat diet and the overweight observed in MAs. It should have been interesting to see the link between obesity and the concentrations of the different OCPs and discuss this aspect.

Minor comments:

  • Line 118, for concentrations it is important to indicate per g of lipids.
  • Line 303, DDT instead of DD.
  • Line 308, cancer instead of cance.
  • Line 309, kidney disease instead of kidney diseas.

Reviewer 3 Report

The paper was well-described concerning the concentration of organochlorine pesticides in the human body.  Each compound is detected by the GC/IDHRMS, but there was no description of each detection limit by the procedures.  Provide detection limits of each compound, if possible.

Reviewer 4 Report

The article " Temporal Trends of Exposure to Organochlorine Pesticides in the U.S.: A Population Study from 2005 to 2016" presented by Mengmeng Li and co-authors is interesting and can be useful to the community studying  chemical residuals and their impact on enviroment and humans . This article agrees with Int. J. Environ. Res. Public Health topics and it is indeed interesting to the readers of this journal. In my opinion, this article should be accepted after minor revision.

Authors started with a good introduction to their research, with sufficient bibliographic research. Their methods are well described. They succeed to present their results and give sufficient comments on several points.

However, there are some minor issues that require attention:

Please check again the page numbering; there are two pages with number “2 of 17” and two pages with number “3 of 17”. Page 4 has no number.

Page 3, Line 99:  “ng/lipid” it is not obvious what do you mean by this? Do you mean “ng of lipid content in the cell”, or ng/g lipid? Please write in a better way.

Page 3, Line 190 and 192: “95%CI” a space should be added after %
